# Ursolic and Oleanolic Acids: Plant Metabolites with Neuroprotective Potential

**DOI:** 10.3390/ijms22094599

**Published:** 2021-04-27

**Authors:** Evelina Gudoityte, Odeta Arandarcikaite, Ingrida Mazeikiene, Vidmantas Bendokas, Julius Liobikas

**Affiliations:** 1Laboratory of Biochemistry, Neuroscience Institute, Lithuanian University of Health Sciences, LT-50161 Kaunas, Lithuania; evelina.gudoityte@gmail.com (E.G.); odeta.arandarcikaite2@lsmuni.lt (O.A.); 2Celignis Limited, Unit 11 Holland Road, Plassey Technology Park Castletroy, County Limerick, Ireland; 3Lithuanian Research Centre for Agriculture and Forestry, Institute of Horticulture, Akademija, LT-58344 Kedainiai Distr., Lithuania; ingrida.mazeikiene@lammc.lt; 4Department of Biochemistry, Medical Academy, Lithuanian University of Health Sciences, LT-50161 Kaunas, Lithuania

**Keywords:** ursolic acid, oleanolic acid, neuroprotection, ischaemia, neurodegeneration, Alzheimer’s disease, Parkinson’s disease, neuro-inflammation, cancer, glioblastoma

## Abstract

Ursolic and oleanolic acids are secondary plant metabolites that are known to be involved in the plant defence system against water loss and pathogens. Nowadays these triterpenoids are also regarded as potential pharmaceutical compounds and there is mounting experimental data that either purified compounds or triterpenoid-enriched plant extracts exert various beneficial effects, including anti-oxidative, anti-inflammatory and anticancer, on model systems of both human or animal origin. Some of those effects have been linked to the ability of ursolic and oleanolic acids to modulate intracellular antioxidant systems and also inflammation and cell death-related pathways. Therefore, our aim was to review current studies on the distribution of ursolic and oleanolic acids in plants, bioavailability and pharmacokinetic properties of these triterpenoids and their derivatives, and to discuss their neuroprotective effects in vitro and in vivo.

## 1. Introduction

The brain is an essential part of the human body, therefore neurological disorders such as neurodegenerative diseases (e.g., Alzheimer’s or Parkinson’s disease), stroke/ischaemia and brain cancer usually have a significant effect on vital functions. Noteworthy, oxidative stress has been recognized as a hallmark of the pathogenesis of these disorders. Moreover, the elevated levels of reactive oxygen (ROS) and nitrogen (RNS) species are implicated in the progression of pathological neuro-inflammation [1]. Therefore, the stimulation of cellular antioxidant systems and quenching of ROS and RNS generation, modulation of cell death and anti-inflammatory pathways have proven to be promising neuro-therapeutic approaches [2,3,4].

Over the last decades the attention to triterpene-enriched plant extracts and especially to purified triterpenoids as bioactive phytochemicals has been considerably increased [5,6,7,8]. Pentacyclic triterpenoids are secondary plant metabolites that arise from cyclization of squalene and are widespread in stem bark and leaves of a variety of plants, and are present in substantial amounts in apple peels [9,10,11,12]. Among various triterpenoids, naturally occurring and synthetic ursane and oleanane triterpenes have been immensely studied due to their wide ranging and promising anti-inflammatory and anticancer activities [13,14,15,16]. Therefore, in this review we will outline the major natural sources and summarize the current studies on the chemistry, bioavailability and pharmacokinetic properties of ursolic and oleanolic acids and some of their derivatives. We will also discuss the most recent findings related to their modulatory effects in neurodegeneration, brain ischaemia and cancer in experimental models in vitro and in vivo.

We retrieved the literature from PubMed, Science Direct, MDPI, Web of Science and Google Scholar. Search terms comprised “ursolic/oleanolic acid”, “triterpene”, “triterpenoid”, “pentacyclic”, “extract”, “plant”, “neuroprotection“, “cancer”, “cell”, “glioma”, “glioblastoma”, “brain”, “ischaemia”, “human”, “rat”, “mouse”, “model”, “bioavailability”, “pharmacokinetics”, “metabolite” and “distribution”. The most recent studies in English with full text available and indexed in Clarivate Analytics Web of Science were included. We also used the reference lists of identified articles to find additional papers. The last search was performed on the 22 February 2021.

## 2. Chemistry, Occurrence and Isolation of Ursolic and Oleanolic Acids

### 2.1. Chemical Structure of Ursolic and Oleanolic Acids

Triterpenes constitute a significant portion of all phytochemicals, and more than 20,000 triterpenoids have been identified so far. Their biosynthesis is based on squalene cyclization, which involves a chair-chair-chair-boat transition state, and results in compounds, comprised of six isoprene units. Such structures are precursors to steroids in both plants and animals [17,18].

Ursolic acid (UA) (3β-hydroxyurs-12-en-28-oic acid) and oleanolic acid (OA) (3β-hydroxyolean-12-en-28-oic acid) are pentacyclic triterpenoids which have therapeutic potential [19]. They are biosynthesized in the mevalonate pathway from the primary sterol metabolism precursor 2,3-oxidosqualene which is transformed into dammarenyl cation by oxidosqualene cyclases. Subsequently, dammarenyl undergoes ring expansion and cyclization to form α-amyrin and β-amyrin skeletons with the characteristic ring system and methyl groups. The α-amyrin constitutes a UA skeleton, whereas its C30 isomer, β-amyrin, forms OA [20]. Thus, UA and OA are structural isomers differing in the position of one methyl group (see Figure 1) and having comparable physicochemical properties and pharmacologic activity [21,22]. UA and OA are ubiquitous in plants and may constitute several percent of the dry weight [17,18]. They occur as free acids or serve as aglycones for triterpene saponins [18].

### 2.2. Natural Sources of Oleanolic and Ursolic Acids

The name OA is derived from the name of the plant species *Olea europaea*, and that plant currently is the main source of commercial OA preparations [24]. OA co-occurs with UA in numerous plant species, including many food, aromatic and medicinal plants [10,17,18,25,26] and UA is usually more abundant than OA (Table 1). Both triterpenoids are often found in the epicuticular waxes and they perform important functions, namely, prevent water loss, serve as a first defence barrier against pathogens and protect from herbivores [26,27,28]. For instance, OA is present as almost pure crystals on the surface of olive leaves and forms a physical barrier against fungal infection [29]. UA possesses surfactant properties and plays a role in the allelopathic effect of plants [30,31].

The synthesis of OA and UA is characteristic of plants of Araliaceae, Asteraceae, Ericaceae, Lamiaceae, Myrtaceae, Oleaceae, Rosaceae, Rubiaceae, Saxifragaceae and Verbenaceae families. Woody or herbaceous plants that belong to these and some other families contain OA and UA in various parts [10,18,25,26]. It has been established that the composition and content of triterpenes may vary between different organs of the same plant. For example, leaves, bark and fruits of *Olea europaea* were found to accumulate different amounts of OA and UA (Table 1). Triterpenes have been shown to be more abundant in the leaves of *Olea europaea* than in the bark: 31 mg/g dry weight (DW) of OA and 3.8 mg/g DW of UA. The bark of olive trees also contained substantial amounts of OA (up to 9.8 mg/g DW) [10]. In contrast, UA predominated over OA in the leaves and bark of *Sambucus nigra* and in medicinal or aromatic herbs of the Lamiaceae family (see Table 1). Moreover, in some plants, detectable amounts of either OA, UA or both, are synthesized in only one organ structure. For instance, OA was present only in the bark of *Betula alba* (up to 11 mg/g DW), and UA was detected only in dry leaves of *Coffea arabica* and *Nerium oleander* (up to 18 mg/g and 12.7 mg/g, respectively) [10]. It is worth noting that the aboveground parts of the herbaceous ornamental peony plant may be used as an alternative source of OA and UA for medicinal purposes. For example, the amount of OA and UA in fresh leaves of different varieties of *Paeonia lactiflora* was 54.26–618.12 μg/g and 36.23–665.14 μg/g, respectively [32].

The quantity of triterpenes in the plant raw material may also depend on the age of the plant. It was shown that *Ocimum basilicum* accumulated 3.6 times more OA in the flowers compared to the roots, and 1.8 to 3.5 times more in older leaves, than in the youngest leaves. Conversely, the level of UA was higher in the youngest leaves of *Ocimum basilicum* when compared to the older leaves. It was also determined that the concentration of UA was 2.7 times higher in the flowers than in the leaves of the plant [33].

As has been already mentioned, plant flowers are also a rich source of triterpenes. High levels of triterpenes, especially UA, were determined in flowers of *Calendula officinalis*, which accumulated up to 20.5 mg/g DW of UA [25]. Different flower parts of *Eriobotrya japonica* were found to contain different amounts of OA and UA. For instance, dry sepals of *E. japonica* contained the highest amount of OA and UA (0.68 mg/g and 3.65 mg/g), while the petals had the lowest contents of OA and UA (0.12 mg/g and 0.60 mg/g) [34].

A large number of triterpenes is characteristic of the fruit peel of the plants in the Rosaceae family. UA and OA have been found to predominate other triterpenes in the cuticular wax of quince, loquat, pear, peach and apple fruits, and UA content was higher than that of OA [26]. Sut et al. recently revealed that UA and OA may constitute up to 79–95% of the total amount of triterpene compounds in apple fruit. Modern commercial apple cultivars can be distinguished by higher amounts of OA (+174%) and UA (+175%) in peels, compared to those of ancient varieties [35]. It has also been shown that the dry peel of *Malus domestica* fruit on average had 9.4 mg/g, *Eriobotrya japonica* had 8.0 mg/g, *Pyrus communis* had 7.25 mg/g, *Chaenomeles japonica* had up to 5.7 mg/g and *Prunus persica* had up to 2.97 mg/g of UA [26]. In addition, the amount of UA in the fruits of ancient apple cultivars has been found to range from 0.20 to 4.20 mg/g and that of OA from 0.24 to 0.87 mg/g. In contrast, the amount of UA in apple fruit of the modern cultivars grown in Lithuanian industrial orchards has been reported to range from 1.18 to 2.58 mg/g, and the amount of OA from 0.32 to 0.47 mg/g [36,37]. Noteworthy, the growth location of apple trees is also relevant to triterpene accumulation. Indeed, apples grown in northern regions were usually found to contain more UA and OA than those grown in the southern locations [38]. Unlike the fruits of Rosaceae plants, persimmon (*Diospyros kaki*) fruits have higher content of OA than UA, 88.57 μg/g fresh weight (FW) and 27.64 μg/g FW, respectively [39]. It is worth adding that berries of *Vitis* spp. and *Vaccinium* spp. can also be used as sources of triterpenes. Dried *Vitis vinifera* berries have been found to contain up to 79.0 mg/100 g of OA, meanwhile, dried cranberries or blueberries contained both OA and UA (17.8 and 65.5 mg/100 g or 13.9 and 11.8 mg/100 g, respectively) [40].

### 2.3. Extraction of Ursolic and Oleanolic Acids from Plants

Methods for the extraction of UA and OA from plants range from the conventional techniques of stirring, maceration and heat reflux extraction to microwave assisted extraction or ultrasound assisted extraction [41,42]. The most prevalent method of extraction of both pentacyclic triterpenes is extraction by immersion in organic solvent, with ethanol, n-butanol, acetone, chloroform or ethyl acetate as the most common solvents [43]. The selection of the most suitable solvent is regarded as the most important step in the preparation of extracts of biologically active compounds. Fu et al. [43] established that the best UA and OA yields from pomegranate flowers can be obtained using chloroform or ethanol—up to 9.2 mg/g of UA and 5.9 mg/g of OA, although ethanol was preferred as a safer alternative. The use of ultrasound resulted in higher yields of UA and OA—12.6 mg/g and 9.7 mg/g, respectively. In addition, the extraction time was reduced to 50 min, compared to 24 h when extraction was performed using stirring or maceration. Cargnin and Gnoatto [19] found that the purest extract having the highest concentration of UA can be obtained by Soxhlet extraction of apple pomace using ethyl acetate as solvent. In this case, UA yield reached up to 3.5% while the use of other solvents (cyclohexane, dichloromethane or methanol) resulted in much lower UA amounts—only up to 1%. Recently, aqueous solutions of surface-active ionic substances have emerged as useful solvents for extraction of hydrophobic substances. Cláudio and colleagues [44] used 1-alkyl-3-methylimidazolium-based ionic liquids combined with several anions for extraction of OA from olive leaves and established that up to 2.5 wt% of OA could be extracted at optimum conditions.

Another critical parameter is the duration of the extraction of UA and OA. Classical methods such as extraction by stirring or maceration usually take 24 h, therefore faster extraction methods, such as microwave assisted extraction or ultrasound assisted extraction, have been developed. The application of microwaves enabled extraction of both UA and OA from the flowers of white dead nettle in 10–30 min, and the extraction efficiency reached 99–100%. Moreover, the efficiency of ultrasound assisted extraction reached 83–91% after 45 min [45]. Fan et al. [46] established that the best UA extraction yield from apple pomace can be obtained after ultrasound assisted extraction for 60 min, without the increase of extraction yield after longer extraction (120 min). Therefore, it can be stated that optimization of the extraction process is crucial in the preparation of UA and OA-enriched plant extracts.

## 3. Bioavailability and Pharmacokinetic Properties of Ursolic and Oleanolic Acids

Pharmacokinetic and cell membrane permeability studies are crucial in clinical development of new biologically active compounds with a view to understanding their behaviour in vivo and to establish an optimal dosage regimen. Pentacyclic triterpenoids generally suffer from low oral bioavailability [47], in particular, UA and OA fall into class IV according to the Biopharmaceutical Classification System due to low aqueous solubility and poor intestinal permeability [20,48]. In line with that, there is large inter and intra-individual absorption variability which presents a challenge of achieving safe and effective drug concentration in the organism [48].

The primary factor affecting bioavailability is the physicochemical properties of the molecule [49]. UA and OA are low molecular weight compounds (456.68 g/mol) [20], containing only three hydrogen bond acceptors and two hydrogen bond donors (see Figure 1). These properties are in accordance with Lipinski’s rule, an estimate of drug likeness [50,51]. However, UA and OA have high lipophilicity (logP_ow_ 6.43 and 6.48, respectively [47]) as well as poor wettability [48,52]. Therefore, their absorption is hindered by poor dissolution and slow partitioning between the cell membrane and extracellular fluid [47,48,50]. It has even been hypothesized that UA may be embedded in phospholipid bilayer but not taken up by the cells [52]. Furthermore, aqueous solubility is aggravated by the crystalline structure of unprocessed UA and OA. Indeed, amorphous state and reduced particle size notably enhanced triterpenoid solubility and dissolution rate [53,54]. Unfavourable characteristics of UA and OA must be addressed in the development of pharmaceutical dosage forms. For instance, solid dispersions of UA in poloxamer 407 [55] or inclusion complexes of OA with cyclodextrins [56] were investigated as opportunities to promote water solubility of these lipophilic triterpenoids.

The second key factor is the ability of the drug to overcome biological barriers [49]. The evidence from in vitro permeability studies implies that passive diffusion is the major mechanism of UA and OA transport [47]. Apparent permeability coefficients, calculated using Caco-2 monolayers, were within the limits of moderate oral absorption [47], and it was suggested that immediate glucuronidation and sulfation in intestinal cells are highly unlikely [57]. However, it has also been reported that UA and OA are substrates of cytochrome P450 enzymes and P-glycoprotein, thus their bioavailability may be restricted by biotransformation and active efflux [48,54,58].

Despite limited absorption, the recovery of intact OA and UA has been reported in animal [53,54,58,59,60,61,62,63,64,65,66] and human [67,68,69] plasma after oral and parenteral administration. Nonetheless, several pharmacokinetic studies revealed that maximal plasma concentration following oral administration of doses up to 300 mg/kg was low (at nanogram quantities) and elimination half-life was relatively short (<1 h) [54,60,63,69]. This pharmacokinetic profile indicates rapid elimination or tissue distribution and suggests that pharmacological effects of UA and OA might not be directly related to plasma concentrations. In fact, it has been shown that UA and OA have wide tissue distribution, as they were detected in the heart, liver, kidney, colon, bladder, brain, spleen, lung, stomach and testis of animals [59,60,70,71]. The liver was found to be the major organ of triterpene disposition [59,60,70], and that is in agreement with established hepatoprotective effects. On the other hand, this may lead to hepatotoxicity, as evidenced by cholestasis after repeated OA administration in mice [72], and to liver-related dose-limiting toxicity in a phase I clinical trial of UA liposomes [73]. It is noteworthy that OA and UA are able to cross the blood-brain barrier, meaning they have the potential to exert neuroprotective effects [70,71]. Targeted delivery to the brain can be achieved by using specific delivery systems, for example, UA nano lipid vesicles in the form of intranasal gel [74].

High lipophilicity predisposes triterpenoids to liver metabolism. A tissue distribution study in mice has demonstrated that while the concentration of UA in plasma was steadily decreasing, the concentration in the liver was increasing (5–240 min) [59]. In addition, the plausibility of liver metabolism has been suggested by the dramatic decrease in OA concentration after the incubation of rat liver microsomes with OA [63]. Oxidation was proposed as a primary pathway of phase I metabolism because hydroxylated derivatives and epoxides were tentatively identified as OA and UA metabolites, respectively [63,75]. When excretion was investigated, unaltered OA and UA were not detected in urine, suggesting non-renal elimination [63]. However, when urine was screened for possible phase II metabolites, renal excretion was confirmed. The metabolites were characterized as glucuronides and sulphates of OA [76] and glucuronic acid, glycine and glutathione conjugates of UA [75]. Hydrophilic conjugates are generally pharmacologically inactive, whereas oxidized metabolites of UA and OA are structurally similar to aromatase inhibitors, thus they could exert anti-estrogenic effects [75]. Overall, appropriate biomarkers and sampling procedures should be selected to accurately evaluate the UA and OA bioavailability with respect to associated health benefits.

## 4. Neuroprotective Effects of Ursolic and Oleanolic Acids

### 4.1. Neuroprotective Effects of Ursolic and Oleanolic Acids in Neurodegeneration

Neurological disorders can be regarded as impairments of the brain or nervous system, resulting in either physical, psychological or both symptoms. Cerebral ischaemia or brain trauma are examples of acute conditions, while gradual memory loss, neurodegenerative diseases or dementia are associated with aging [77]. Proper brain function is dependent on neuronal signal transduction and supportive activity of glial cells. The brain is also characterised by the highest metabolic rate among all organs related to the intense ROS or RNS formation and consequently the requirement of an effective antioxidant system [77]. The main antioxidative enzymes in the brain are glutathione peroxidase (GP), superoxide dismutase (SOD), catalase (CAT), peroxidase, haem oxygenase, quinone oxidoreductase 1 and γ-glutamylcysteine ligase. The expression of various antioxidant enzymes is regulated by nuclear factor erythroid 2-related factor 2 (Nrf2), an important transcription factor involved in the maintenance of redox and metabolic homeostasis. It is also known that non-enzymatic glutathione plays a critical role in ROS scavenging [77]. Therefore, antioxidant properties are important for potential therapeutic agents. UA has been shown to possess antiradical activity in vitro. Salau et al. [3] found that UA was more potent than ascorbic acid as reflected by IC_50_ values obtained by DPPH (2,2-diphenyl-1-picrylhydrazyl) radical scavenging potential (2.08 μg/mL and 7.64 μg/mL, respectively) and ferric reducing antioxidant power (FRAP) (0.75 μg/mL and 20.17 μg/mL, respectively) assays. Although the difference may arise from the distinction of hydrophilicity and the selectivity of the assays [78], it strongly suggests the antioxidant potential of UA. Moreover, the pre-treatment with UA has abolished kainate-induced free radical generation in primary culture of hippocampal neurons [79]. Antioxidant effects of UA have been demonstrated in a wide range of experimental models (see Table 2), both in vitro and in vivo, with single or repeated administration. UA has been shown to increase the activity of CAT [80,81,82], SOD [3,77,79,81], glutathione (GSH) [3,80,81,82], GP [78] and to activate the Nrf2-pathway [83,84]. The final result of the majority of experiments was the reduction of lipid peroxidation expressed as the decrease in the level of malondialdehyde (MDA) [3,80,81,82,83,84]. Ex vivo UA has been also able to decrease the activity of α-chymotrypsin, which is known as the marker of oxidative injury [3]. OA has been found to elicit antioxidant effects in a similar manner to UA. It induced the reduction of intracellular ROS levels in vitro [85], and it was demonstrated in vivo that OA activated GSH and SOD as well as decreased the level of MDA [16,86] (see Table 2).

Neurodegenerative diseases (see also Table 2: Parkinson’s or multiple sclerosis experimental models) are associated with neuro-inflammation—a complex process, regulated by microglia and astrocytes [87]. These cells produce pro-inflammatory factors such as tumour necrosis factor alpha (TNF-α) [88]. Under certain conditions TNF-α can promote inflammation by activation of nuclear factor-κB (NF-κB) and mitogen-activated protein kinase (MAPK) signalling pathways and induce apoptotic processes [89]. Therefore, cytokines and transcription factors are feasible targets for anti-inflammatory therapy and the effects of triterpenoids have received considerable attention. It has been shown that prolonged administration of UA can lead to downregulation of the NF-κB pathway [82,84,90], possibly resulting in decreased levels of interleukins IL-1β [90], IL-12 [82], IL-6 [90], interferon gamma (IFN-γ) [82], matrix metalloproteinase-2 and 9 (MMP-2, MMP-9) [91]. Moreover, UA can promote the expression of genes encoding anti-inflammatory cytokines IL-4 and IL-10 [82]. The anti-inflammatory potential of OA has been also demonstrated. For example, OA could decrease the expression levels of TNF-α, IL-1β and IL-6 in BV2 cells [16].

As UA and OA are able to modulate various anti-inflammatory pathways, they can also have neuroprotective effects. Indeed, it has been reported that UA increased the myelinated area, oligodendrocyte count and myelin basic protein (MBP) content in a multiple sclerosis mouse model after administration for six weeks, as UA acted as an agonist of peroxisome proliferator activated receptor γ (PPARγ) [92,93]. It is known that PPAR-dependent transcription factors play a crucial role in the inflammatory response of the CNS by inhibition of NF-κB [94] and downregulation of genes encoding for pro-inflammatory proteins such as cyclooxygenase-2 (COX-2), MMP-9 [91], scavenger receptor A, inducible nitric oxide synthase (iNOS), as well as the inhibition of the synthesis of pro-inflammatory cytokines [95,96,97,98]. It is well documented that NO in the brain is generated by neuronal nitric oxide synthase (nNOS), whereas inflammation triggers the activation of iNOS [87]. It has been shown that prolonged administration of UA resulted in the reduced activity of iNOS [77,89]. Similarly, OA reduced the level of NO in BV2 cells, and this was associated with the downregulation of the expression of the iNOS encoding gene [16]. Given the multiple interrelated molecular pathways described above, the protective effects of UA and OA can be ascribed to antioxidative and anti-inflammatory activity. Furthermore, the inhibition of apoptosis could also contribute to neuroprotection. Studies using different experimental models indicated that UA could decrease the level of apoptosis effectors, such as caspase 3 [80,99] and caspase 9 [80]. The oxygen-glucose deprivation (OGD) experiments with organotypic hippocampus slices have also demonstrated the protective effect of UA evidenced by modulation of gene expression via AKT/mTOR/HIF-1α pathway, resulting in the increased level of Bcl-2 and the decreased level of Bad [99].

It is noteworthy that the underlying target of UA and OA might be mitochondria, organelles that are involved in both oxidative stress and neuro-inflammation [87]. Recently it has been shown that prolonged administration of UA affected the functionality of mitochondrial electron transport chain. In brain mitochondria, UA improved the enzymatic activity of mitochondrial complex I in rotenone-induced Parkinson’s disease model in vivo and increased the expression of mitochondrially encoded cytochrome c oxidase 1 (MrCO1) [81]. Mitochondrial membrane potential has also been found to be sustained by UA (at 10 μM concentration for 10 min) in an excitotoxicity model of hippocampal neurons [79]. In another study, it was revealed that after the treatment of PC12 cells with OA, H_2_O_2_-reduced mitochondrial membrane potential was restored and the activity of succinate dehydrogenase (SDH) was improved [86]. It is worth mentioning that UA at 1–10 µM concentration had no effect on isolated mouse brain mitochondrial respiration at different metabolic states [100] (see also Appendix A). However, it was previously observed that UA had uncoupling and antioxidant effects on rat heart mitochondria [101]. Thus, the relationship between the administration of UA or OA and the favourable effects on mitochondrial functions under oxidative stress, ischaemic damage or neuro-inflammation requires further investigation as observed outcomes might depend on experimental models.

**Table 2 ijms-22-04599-t002:** Effects of UA and OA on experimental models of brain pathologies.

**Ursolic Acid**				
**Brain** **Pathology**	**Experimental Model**	**Dosage**	**Beneficial Effects**	**Reference**
Trauma/ischaemic damage	1 h MCAO with 24 h reperfusion. Neonatal rat hippocampal slices after oxygen-glucose deprivation.	10, 50 and 100 mg/kg.	Apoptosis ↓, Protection via AKT/mTOR/HIF-1α pathway, Bcl-2 ↑, Bad ↓ and Caspase 3 ↓	[99]
	2 h MCAO and 48 h reperfusion.	Post-conditioning with 5, 10 or 20 mg/kg.	MMP-2 and MMP-9 ↓, TIMP1 ↑, PPARγ-positive cells ↑	[91]
	MCAO rat model with 24 h reperfusion.	Post-conditioning with 130 mg/kg.	Nrf2 pathway activation, TLR4 ↓ and NF-kB ↓, MDA ↓	[84]
	Mouse traumatic brain injury after 24 h.	Pre-conditioning with 100 mg/kg.	GP ↑, SOD ↑ and MDA ↓, apoptosis ↓ via Nrf2-ARE signalling pathway.	[83]
	Subarachnoid haemorrhage rat model by endovascular perforation.	Post-treatment 50 mg/kg.	SOD ↑, CAT ↑, GSH and GSSH ↑, MDA ↓, caspase-3 and-9 ↓	[80]
Excitotoxicity	Primary neuronal cultures from the hippocampus of 7-day-old rats were treated with 150 mM kainate for 2 h.	Pre-treatment for 10 min with 5, 10 or 15 μM.	Non-NMDA receptor modulation, intracellular ROS ↓, mitochondrial membrane potential stabilisation.	[79]
Inflammation	To mimic NF-κB pathway mice were subcutaneously injected with D-galactose.	10 mg/kg/d for 8 weeks.	COX-2 ↓, iNOS ↓, IL-1β ↓, IL-6 ↓, TNF-α ↓, ROS ↓, advanced glycation end products ↓	[90]
Multiple sclerosis	Multiple sclerosis mouse model.	Prolonged oral administration of 25 mg/kg.	MBP+ ↑, myelinated axons ↑, PPARγ pathway activation.	[93]
	Multiple sclerosis mouse model.	Daily use of drinking water with 1 mg/mL for 6 weeks.	myelinated area in corpus callosum ↑, MBP ↑	[92]
Parkinson’s disease	Parkinson’s disease model established by rotenone infusions	30-day administration of 5 and 10 mg/kg.	CAT ↑, SOD ↑, GSH ↑, MDA ↓, TNF- α ↓, improved mitochondria complex I enzymatic activity, MtCO1 gene expression ↑, tyrosine hydroxylase positive neurons ↑, glial fibrillary acidic protein ↓	[81]
	Parkinson’s disease rat model designed by intraperitoneal injections of 1-methyl-4-phenyl-1,2,3,6-tetrahydropyridine	Orally 25 mg/kg for 21 d.	CAT ↑, GSH ↑, MDA ↓, tyrosine hydroxylase-positive dopaminergic neurons ↑, NF-κB ↓, TNF-α ↓, IFN-γ ↓, IL-12 ↓, IL-10 ↑, IL-4 ↑	[82]
**Oleanolic Acid**				
**Brain** **Pathology**	**Experimental Model**	**Dosage**	**Beneficial Effects**	**Reference**
Inflammation	Mouse microglial BV2 cell line activated by lipopolysaccharide.	Pre-treatment with 0.5–25 µM.	IL-1β ↓ , IL-6 ↓, TNF-α ↓, NO ↓, GSH ↑, iNOS ↑	[16]
Parkinson’s disease	PC12 cell culture treated with 6- Hydroxydopamine.	Pre-treatment and post-treatment with 100 mg/kg.	Dopamine ↑, intracellular ROS ↓, neuronal cell survival ↑	[85]
Ischaemic damage	Wistar rat focal cortical hypoxia induced by cobalt chloride injection.	Intraperitoneal injection with 6 mg/kg/d for 7 d.	Neuronal survival ↑, dendrite recovery ↑, astroglial and microglial reaction ↓.	[102]
	Bilateral common carotid artery ligation in mice, and PC12 cells pre-treated with H_2_O_2_	Pre-administration of 50 and 25 mg/kg, respectively.	Infarct zone size ↓, mitochondrial membrane potential ↑ and succinic dehydrogenase ↑, SOD ↑ and GP ↑, MDA ↓	[86]

↑ indicates stimulation of a process or an increased level of compound, ↓ indicates reduction of process activity or a decreased level of compound; MCAO—experimental middle cerebral artery occlusion, SOD—superoxide dismutase, CAT—catalase, GSH—glutathione, GSSH—oxidized glutathione, GP—glutathione peroxidase, Nrf2—nuclear factor erythroid 2-related factor 2, TLR4—toll-like receptor 4, MDA—malondialdehyde, MBP—myelin basic protein, IFN-γ—interferon gamma, IL—interleukin, TNF-α—tumour necrosis factor alpha, NO—nitric oxide, iNOS—inducible nitric oxide synthase, MMP—matrix metalloproteinase, PPARγ—peroxisome proliferator activated receptor γ.

### 4.2. Glioblastoma and Ursolic/Oleanolic Acids

Glioblastoma is the most common and the most aggressive type of primary brain tumour, accounting for approximately 55% of gliomas [103,104]. The survival of patients with glioblastoma that have undergone standard treatment encompassing surgical resection and radiation therapy followed by chemotherapy with temozolomide (TMZ) remains within 14–18 months [105,106]. More importantly, tumour relapse occurs in almost all patients, and in such cases, glioblastoma often becomes resistant to chemotherapy [107]. Therefore, novel agents and therapies for the treatment of glioblastoma are needed, as newly developed agents have failed to outmatch TMZ so far [108].

Recently, in a number of reviews UA/OA-containing plant extracts, purified compounds and their natural or chemically synthesized derivatives have been reported to possess anti-tumour activity in skin, breast, lung, gastric, liver, intestine, prostate and pancreatic cancer models in vitro and in vivo with non-toxic or minimal inhibitory effect on normal cells [14,109,110,111,112,113,114]. Moreover, new data about the potential of these pentacyclic triterpenoids against glioblastoma have also emerged [15]. Byun et al. [115] have recently isolated several C-27-carboxylated OA derivatives (C27OAs) from the dried roots of *Astilbe rivularis*, and showed that 3β-hydroxyolean-12-en-27-oic acid, 3β,6β,7α-trihydroxyolean-12-en-27-oic acid and 3β-*trans*-*p*-coumaroyloxy-olean-12-en-27-oic acid at 10 µM concentration could exclusively sensitize the extrinsic apoptotic pathway via the p38 MAPK and CHOP-mediated DR5 expression without affecting the intrinsic pathway in human LN-428 and U-251 MG glioblastoma cell lines in vitro. The results suggest that certain C27OAs might be developed as specific TNF-related apoptosis-inducing ligand (TRAIL) sensitizers and used as chemotherapy agents in glioblastoma patients with low levels of caspase-8 or p53 activity. In another study [116], synthetic OA derivatives, C-28 methyl ester (CDDO-Me) and C-28 imidazole (CDDO-Im) of 2-cyano-3,12-dioxooleana-1,9(11)-dien-28-oic acid, have been shown to exhibit potent apoptosis-inducing activity in human U-87 MG and U-251 MG glioblastoma cells. It is interesting that the tested compounds at 2.5–10 µM concentration induced apoptotic cell death through both extrinsic and intrinsic pathways, as the activation of pro-caspases-3, 8 and 9, mitochondrial depolarization and the release of cytochrome c from mitochondria were observed. Moreover, CDDO-Me inhibited the expression of anti-apoptotic and pro-survival signalling molecules (p-Akt, p65 and Notch1) in the same cell lines. In addition, OA has been also investigated for its ability to affect metabolic activity, viability and cell cycle of U-87 cells in vitro [117]. It was found that after 24 h of treatment OA at 100 µg/mL (approx. 219 µM) induced both apoptosis and necrosis of cancer cells almost to the same extent, and the IC_50_ value was determined as 163.6 µg/mL (approx. 358 µM). In addition, OA-treated glioblastoma cells exhibited the increase in expression levels of proteins involved in MAPK signalling, and cell cycle arrest at G1 phase.

UA has been also found to exert a cytotoxic effect in human U-251 MG glioblastoma cells with an IC_50_ value as low as 20 µM [118]. The treatment induced JNK-dependent, caspase-independent and lysosomal associated mechanism of cell death that resulted in rapid mitochondrial membrane depolarisation. In addition, UA demonstrated greater cytotoxicity than conventional chemotherapeutics like TMZ in spite of inactivity towards O-6-methylguanine-DNA methyltransferase (MGMT), which is a known player in TMZ-resistant glioblastomas. UA (at 20 µM) has been also found to induce apoptosis in U-251 cells by suppressing TGF-β1 signalling pathway, thus revealing an alternative mechanism of anti-cancer activity [119]. In other in vitro glioblastoma models, UA (at 17.5 µM) has been shown to induce necrosis in TMZ-resistant human DBTRG-05MG cells through mitochondrial permeability transition pore opening and ATP level decline [120] or at 40 µM concentration UA treatment has led to cell cycle arrest at G1 phase, endoplasmic reticulum stress-induced JNK activation and autophagy in U-87 MG cells [121]. However, UA from *Chamaenerion angustifolium* at 50 µM concentration induced both apoptosis and necrosis in U-87 MG cells, and this effect was explained by UA interaction with the PI3K/Akt signalling pathway as predicted by molecular docking [122]. UA (at 12.5 µM) has also enhanced the cytotoxicity of TMZ in human LN-18 and T98G glioblastoma cells by downregulating MGMT expression, and at 50 mg/kg UA potentiated the efficacy of TMZ in BALB/c mice with LN-18 xenograft [123]. Furthermore, UA from *Rosmarinus officinalis* at 20 µM has reduced IL-1β or TNF-α-induced rat C6 glioma cell invasion in transwell chambers and inhibited the enzymatic activity and expression levels of MMP-9 via the blockage of the NF-kB-dependent pathway [124]. UA from the methanolic extract of the leaves of *Sarauja roxburghii* at 100 µM concentration has also exhibited cytotoxicity against C6 cells in vitro [125]. Bergamin et al. [126] found that UA at 15–20 µM increased the number of C6 glioma cells in sub-G1 phase, induced apoptotic cell death and also reduced the expression level and activity of protein kinase B (Akt) in vitro. However, 15 mg/kg/d (for 10 d) of UA did not affect the tumour size in an orthotopic glioma model in vivo [126]. In another study utilizing an in vivo model of glioblastoma, the anti-tumour effects of triterpenoids have been established [127]. When C6 tumour-bearing rats were given OA solution by gavage (40 mg daily for 7 d) and underwent irradiation therapy, the combined effect of OA and radiation was demonstrated, which resulted in the decreased growth rates of tumours and the prolonged survival period of tumour-bearing rats.

It is noteworthy that UA (20–100 µM, 48 h of incubation) suppressed metabolic activity, decreased ATP content and increased the number of necrotic cells in both rat C6 glioma and primary mouse astrocyte cell cultures to the same extent [128]. Thus, it is obvious that the observed OA and UA anti-tumour effects are rather controversial, and additional research is certainly needed to clarify the molecular mechanism of action and neuroprotective potential of these pentacyclic triterpenes.

## 5. Conclusions

In conclusion, various parts of medicinal or aromatic plants, and also the pulps and peels of many fresh fruits, are sources of structurally isomeric pentacyclic triterpenoids UA and OA. Moreover, within the last several years UA/OA-enriched extracts, purified compounds and their synthetic derivatives have become subjects of increasing attention due to their potential anti-inflammatory and anticancer effects. Thus, the reviewed data suggest that neuroprotective activity of UA/OA and their derivatives could be associated with their ability to stimulate the cellular antioxidant defence systems and to downregulate the pro-inflammatory pathways in neurodegeneration or ischaemic brain damage models in vitro and in vivo. Moreover, UA/OA have also demonstrated anticancer activity, which might be related to the potency of these pentacyclic triterpenoids to activate or sensitize intracellular pathways leading to cell death, and to suppress tumour cell growth, proliferation and migration. However, further research is certainly needed to clarify the mechanism of action, since there is obvious variation in experimental data depending on the treatment regimen and applied models. In addition, the chemical modifications of UA and OA, as well as novel drug delivery systems, are subjects for future research aimed at improving the bioavailability and effectiveness of original compounds.

## Figures and Tables

**Figure 1 ijms-22-04599-f001:**
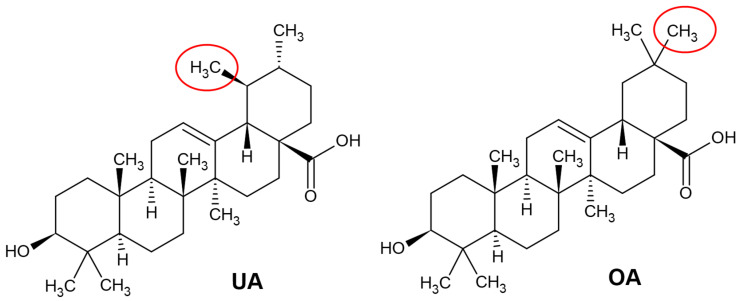
Chemical structures of ursolic acid (UA) and oleanolic acid (OA) (drawn according to [23]). The structural differences between these two isomers are marked in red ovals.

**Table 1 ijms-22-04599-t001:** Natural sources of OA and UA mg/g DW (dry weight).

Plant Species	Family	Plant Part	OA mg/g DW	UA mg/g DW	Reference
*Betula alba*	Betulaceae	bark	11.0	not detected	[10]
*Calendula officinalis*	Compositae	flowers	not detected	20.5	[25]
*Chaenomeles japonica*	Rosaceae	fruit peel	not detected	5.7	[26]
*Coffea arabica*	Rubiaceae	leaves	not detected	18.0	[10]
*Crataegus pinnatifida*	Rosaceae	leaves	1.0	5.2	[10]
*Eriobotrya japonica*	Rosaceae	fruit peel	not detected	8.0	[26]
		flowers	0.9	3.6	[34]
*Lavandula angustifolia*	Lamiaceae	herbs	4.5	15.9	[10]
*Ligustrum lucidum*	Oleaceae	leaves	6.3	9.8	[41]
*Malus domestica*	Rosaceae	fruit peel		9.4	[26]
*Melissa officinalis*	Lamiaceae	herbs	1.6	6.7	[10]
*Nerium oleander*	Apocynaceae	leaves	3.7	12.7	[10]
*Ocimum basilicum*	Lamiaceae	herbs	not detected	3.0	[10]
*Olea europaea*	Oleaceae	leaves	31.0	3.8	[10]
		fruits	21.0	not detected	[10]
		bark	9.8	not detected	[10]
*Origanum majorana*	Lamiaceae	herbs	1.9	6.6	[10]
*Origanum vulgare*	Lamiaceae	herbs	not detected	2.8	[10]
*Panax quinquefolium*	Araliaceae	roots	3.1	not detected	[25]
*Prunus persica*	Rosaceae	fruit peel	not detected	3.0	[26]
*Pyrus communis*	Rosaceae	fruit peel	not detected	7.2	[26]
*Salvia officinalis*	Lamiaceae	herbs	6.7	18	[10]
*Sambucus nigra*	Adoxaceae	leaves	1.2	5.8	[10]
		bark	0.8	3.2	[10]
*Satureja montana*	Lamiaceae	herbs	1.4	4.9	[10]
*Silphium trifoliatum*	Asteraceae	leaves	22.0	15.5	[25]
*Thymus vulgaris*	Lamiaceae	herbs	3.7	9.4	[10]

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
