# Peer review of "Ursolic and Oleanolic Acids: Plant Metabolites with Neuroprotective Potential"

_ijms, 2021, doi:10.3390/ijms22094599_

Round 1

Reviewer 1 Report

This review Gudoityte et al. represents an important contribution to the field. The work is clearly planned, organized, and nicely divided into relevant chapters. Some English corrections are needed.

Author Response

Thank you for your opinion. We have revised the manuscript keeping British English throughout the text.

Reviewer 2 Report

The manuscript in reference compiles the up-to-date information on the reported neuroprotective effects of two pentacyclic triterpenic acids, ursolic acid (UA) and eleanolic acid (OA). The manuscript is interesting, has a good writing level, and has important information. However, I consider that the manuscript suffers from a lack of information that should be included to supplement and provide a broad view of these naturally-occurring compounds and their neuroprotective effects.

Major points

  1. It is attempted that the manuscript deals with neuromodulatory effects of UA and OA, but in the text, specifically in section 4, topics on neuroprotection (during neurodegeneration and glioblastoma events) are mentioned, which I consider very restricted because it is not discussed particularly neuromodulation. Neuroprotection is not, strictly speaking, similar to neuromodulation. In fact, effects on neurotransmission and synaptic events are not included. I suggest checking whether it is appropriate to use the term neuromodulatory in the given context of the present review and title.
  2. The information on biosynthesis remains to be expanded (highly summarized in lines 53-55), specifying even the most-important biosynthetic details that originate the structural difference of these triterpenic acids, based on the methyl group position at ring E.
  3. A section about chemical analysis of these triterpenic acids is missing, after section 2.3, especially with commonly used techniques, such as LC-DAD, LC-MS / MS and NMR, specifying optimal conditions and pertinent examples. Also include information on practical details related to separation and/or purification techniques for these two compounds, e.g., classical or flash chromatography, countercurrent, etc.
  4. Information regarding toxicity studies in cells, animals and humans of these two triterpenes is also missing. It would even be important to mention something about excitotoxicity if applicable.
  5. Additionally, clarify if there is any information on clinical trials of these compounds within the central topic of the review, i.e., neuroprotective effects?
  6. And finally, I am not convinced that the review includes information and data that have not been previously published, because this document is a narrative review of scientific literature, which is already published and peer-reviewed. Including unpublished information that has not been peer-reviewed within a scientific writing structure, including methodological details, results and discussion, and conclusions, is not correct because that information would be validated herein without having the respective support from the previous peer-review process. Accordingly, I suggest not including those non-published data/information generated by authors, specifically in lines 301-303 and 386-392.

Minor points

  1. Figure 1. Absolute configuration on various carbons (e.g., 3, 8, 10, 17, 18, 19, 24) are missing. In the case there are several options, I recommend indicating which are the most common.
  2. Table 1. Clarify if the listed sources are the most common and/or abundant ones, since these two triterpenes, as indicated in line 61, are very ubiquitous in plants and animals. In addition, indicate in this same table, for the columns OA mg/g DW and UA mg/g DW, what is the meaning of the blank spaces (e.g., not measured, not reported, or not detected). It is important to clarify these details in this table 1.
  3. Lines 239-242: I suggest including in the text a reasonable structural explanation that allows to rationalize the fact that UA is a more potent antioxidant than ascorbic acid.
  4. Lines 291-308: Since it is suggested that the target of OA/UA is mitochondria, is there any information on synaptic effects dependent on, e.g., metabolic demand, ATP stores preservation, ion channels regulation, glutamatergic transmission, and excitotoxicity protection? It is important to include information about it.
  5. Lines 408-409: Has some study been carried out on structural modifications of UA and OA to improve effectiveness and bioavailability? It would be important to include this info in this review.

Reviewer 3 Report

This is a very interesting review. They are two chemical compounds of great interest due to their biological activity and their industrial applications. The text is well written and easy to read and is interesting to IJMS readers. Some aspects that I think should be corrected or modified: 1. I consider that the objective of the work (“to review the current knowledge about the distribution of ursolic and oleanolic acids in plants, bioavailability and pharmacokinetic properties of these triterpenoids and their derivatives, and to discuss their neuromodulatory effects in vitro and in vivo”) it is too ambitious. The number of species that have these compounds is very large, and to be able to treat them would require a much more extensive text than the one used by the authors of this manuscript. Therefore, I consider that the objective and title of the manuscript should be modified, for example “Approach to the knowledge of ursolic and oleic acids ………). 2. Include a Material and Method section to find out how the bibliographic review has been carried out. This section is essential since we must know the databases reviewed and the keywords used. It would be interesting if the authors had discussed in some section the traditional knowledge of at least some taxon of each family that has these chemical compounds. Many of these traditional uses are justified by the biological activity of ursolic and oleic acids.

Reviewer 4 Report

This review article is about the potential of ursolic and oleanolic acid in the treatment of neurodegenerative disorders and brain-related cancers. The structure is well organized describing the natural occurence of the above mentioned natural compounds, all biological activities that may be involved in neuroprotection and a number of references are directly related to the research on neurodegeneration and neural cancers. 

I am really glad that such review has been written since this topic has not been previously reviewed and this article gives a nice overview and suggestions to all scientists involved in the medicinal chemistry of triterpenes, especially those that may be interested in neural diseases. This article is going to fill the gap among reviews on hepato- and nephro-protective actions of oleanolic and ursolic acids.

From above mentioned reasons, I reccomend the article to be published as is, in my opinion, no changes need to be done. 

Author Response

Thank you for your positive and encouraging opinion.

Round 2

Reviewer 2 Report

Authors addressed adequately my suggestions and comments, manuscript improved in clarity and quality, so I recommend acceptance of this manuscript in this current form.

Author Response

(The authors gave the same response as above.)

Reviewer 3 Report

The authors have heeded some of the other authors' suggestions and also some of those that he proposed. That is why the manuscript has improved remarkably.

I was negatively surprised by the response to my suggestion that the authors discuss in some section the traditional knowledge of at least some taxon of each family that has these chemical compounds. I cannot conceive that a researcher who works in phytochemistry does not know anything about traditional uses of medicinal plants, and even less that he does not know how to locate that information, for such well-known species as Lavandula angustitolia, Thymus vulgaris, Origanum sp, Olea Europea, or many others mentioned. in the text….

On the other hand, I still consider the title too ambitious considering the number of plants that have these chemical components.

Author Response

In this review we focused on the ursolic and oleanolic acids as ubiquitous phytochemicals and methods for their extraction from plant materials. Therefore, the proposed topic on traditional use of certain medicinal plants would be a little bit out of scope of the Special Issue and our manuscript. Our hesitation is based on the following: 1. A list of medicinal and well-known herbs depends on a region and certain traditional use (Eastern, Western,..), therefore it will always be a feeling that some plants should be included into the analysis or excluded from the list; 2. During the preparation of our manuscript we followed a requirement “Please note that it is necessary to clarify the exact functional ingredient in the research paper, paper only on mixed extraction is not fit for our journal.” (https://www.mdpi.com/journal/ijms/special_issues/Neuromodulatory_Plant). Thus, our search on the use of extracts as neuroprotective agents (see below the References) showed that the crucial information, i.e. chemical composition was missing, and it was impossible to relate the observed beneficial effects to the unidentified ursolic or oleanic acids.

In addition, we would like to note a positive feedback from other three referees, thus we believe the present title clearly presents the main idea.

References:

https://www.mdpi.com/2076-3921/9/9/806/htm

https://www.sciencedirect.com/science/article/abs/pii/S0944711312002000?via%3Dihub

https://www.ncbi.nlm.nih.gov/pmc/articles/PMC4146066/

https://www.sciencedirect.com/science/article/abs/pii/S0944711310001893?via%3Dihub

https://link.springer.com/article/10.1007/s00580-015-2070-7

https://pubmed.ncbi.nlm.nih.gov/31475590/